

# Investigating altered brain functional hubs and causal connectivity in coronary artery disease with cognitive impairment

Rui Qin[1], Tong Li[1], Cuicui Li[1], Lin Li[1], Ximing Wang[1] and Li Wang[2]

[1] Department of Radiology, Shandong Provincial Hospital affiliated to Shandong First Medical University, Jinan, Shandong Province, China
[2] Department of Health Management Center, Shandong Provincial Hospital affiliated to Shandong First Medical University, Jinan, Shandong Province, China

Corresponding authors
Ximing Wang, wxming369@163.com
Li Wang, wli6767@126.com

## ABSTRACT

**Background:** Coronary artery disease (CAD) and cognitive impairment (CI) have become significant global disease and medical burdens. There have been several reports documenting the alterations in regional brain function and their correlation with CI in CAD patients. However, there is limited research on the changes in brain network connectivity in CAD patients. To investigate the resting-state connectivity and further understand the effective connectivity strength and directionality in patients with CAD, we utilized degree centrality (DC) and spectral dynamic causal modeling (spDCM) to detect functional hubs in the whole brain network, followed by an analysis of directional connections. Using the aforementioned approaches, it is possible to investigate the hub regions and aberrant connections underlying the altered brain function in CAD patients, providing neuroimaging evidence for the cognitive decline in patients with coronary artery disease.

**Materials and Methods:** This study was prospectively conducted involving 24 patients diagnosed with CAD and 24 healthy controls (HC) who were matched in terms of age, gender, and education. Functional MRI (fMRI) scans were utilized to investigate brain activity in these individuals. Neuropsychological examinations were performed on all participants. DC analysis and spDCM were employed to investigate abnormal brain networks in patients with CAD. Additionally, the association between effective connectivity strength and cognitive function in patients with CAD was examined based on the aforementioned results.

**Results:** By assessing cognitive functions, we discovered that patients with CAD exhibited notably lower cognitive function compared to the HC group. By utilizing DC analysis and spDCM, we observed significant reductions in DC values within the left parahippocampal cortex (PHC) and the left medial temporal gyrus (MTG) in CAD patients when compared to the control group. In terms of effective connectivity, we observed the absence of positive connectivity between the right superior frontal gyrus (SFG) and PHC in CAD patients. Moreover, there was an increase in negative connectivity from PHC and MTG to SFG, along with a decrease in the strength of positive connectivity between PHC and MTG. Furthermore, we identified a noteworthy positive correlation (r = 0.491, $p$ = 0.015) between the strength of connectivity between the PHC and the MTG and cognitive function in CAD patients.
**Conclusions:** These research findings suggest that alterations in the connectivity of the brain networks involving SFG, PHC, and MTG in CAD patients may mediate changes in cognitive function.

## INTRODUCTION

Coronary artery disease (CAD) refers to stenosis and occlusion of the coronary arteries caused by atherosclerosis, which can lead to myocardial ischemia, hypoxia, and consequently impaired cardiac function. As the global population ages and lifestyle habits change, CAD is receiving increasing attention worldwide (*Camici & Crea, 2007*). Cognitive impairment (CI) refers to the declining state of cognitive function prior to dementia, with the core symptoms being impaired attention, memory, language, and so on. CI not only has a negative impact on patients' lives but also increases the burden on families and society. With the continuous development of CAD research, an increasing number of studies suggest a link between CAD and CI (*Gross et al., 2019*; *Lo et al., 2019*). Furthermore, it has been established that a previous diagnosis of CAD is an autonomous risk factor for CI (*Balbaid et al., 2020*). Currently, potential underlying mechanisms between CAD and cognitive function may include chronic cerebral hypoperfusion, chronic inflammation, and other factors (*Kernan et al., 2014*; *Shabir, Berwick & Francis, 2018*).

Magnetic resonance imaging (MRI) is a noninvasive imaging technique that plays a crucial role in investigating the neuroimaging mechanisms associated with CI in individuals with CAD. Currently, there is still limited research on brain MRI in the field of CAD, mainly focused on brain structural imaging and local brain function research. In terms of brain structural imaging research, *Yokoyama et al. (2022)* evaluated 356 CAD patients' brain MRI and found a higher prevalence of cerebral infarction and cerebral hemorrhage in CAD patients. Some studies have reported significant reductions in gray matter volumes in various brain regions, including the superior frontal gyrus, inferior frontal gyrus, superior temporal gyrus, superior parietal gyrus, inferior parietal gyrus, and posterior cerebellar gyrus, in CAD patients. These reductions were found to be positively correlated with CI in CAD patients (*Anazodo et al., 2013*). Additionally, functional brain imaging studies have demonstrated lower amplitude of low-frequency fluctuations (ALFF) values in the right precuneus, left supramarginal gyrus, left angular gyrus, and left cingulate gyrus in CAD patients compared to healthy controls (HC). Moreover, lower ALFF values in CAD patients have been associated with lower scores on the Montreal Cognitive Assessment (MoCA) and visual impairments (*Zhang et al., 2022*). These findings suggest that both the structure and function of the brain are altered in CAD patients compared to healthy individuals, although current research on changes in brain functional networks and their effects on CAD patients is limited to localized brain regions.

With advancements in MRI technology, a plethora of analytical methods have emerged for studying the intricate connectivity of the entire brain network. Degree centrality (DC)

is a common node centrality index used in network analysis that can measure the degree of connectivity between a node and other nodes, revealing the connectivity of a node, and can evaluate network centrality without the need for regions of interest (*Zuo & Xing, 2014*). Unlike functional connectivity, spectral dynamic causal modeling (spDCM) focuses on "effective connectivity (EC)," which refers to the directional influence of one brain region on another. By modeling the flow of information, spDCM provides insights into how one region influences another (*Friston, Harrison & Penny, 2003*). By approximating the model in the frequency domain, this method reduces the complexity of model estimation, and both the stability and accuracy of the results are good (*Razi et al., 2015*). Currently, spDCM has been applied in other diseases, but it has not been used in patients with CAD.

In patients with CAD, cognitive decline may be attributed to abnormalities in the functional connectivity of the brain network. To better understand these abnormalities and unravel the complex structure of the brain's functional network, we conducted this study using DC analysis. This method allowed us to identify brain regions with abnormal functional connectivity across the entire network. To delve deeper into the understanding of these alterations, we employed spDCM to explore the causal connections and underlying mechanisms driving these changes. Our hypothesis was centered around the idea that the directional connectivity patterns of brain nodes, as indicated by DC analysis, could provide valuable insights into the central neural mechanisms associated with CAD characteristics.

# MATERIALS AND METHODS

This observational cross-sectional study was prospectively conducted at Shandong Provincial Hospital affiliated with Shandong First Medical University from September 2022 to June 2023. The inclusion criteria for CAD patients were as follows: (1) aged between 35 and 75 years; (2) clinically diagnosed with CAD and confirmed by imaging examination; (3) able to complete magnetic resonance imaging scans. The exclusion criteria were as follows: (1) patients with cerebral infarction or carotid artery stenosis or occlusion; (2) patients with contraindications for MRI; (3) patients with consciousness or mental disorders. The control group consisted of volunteers who had undergone detailed cardiovascular examinations in our hospital and had no neurological or cardiovascular diseases. All participants in this study underwent cardiac coronary computed tomography angiography examination to assess the condition of their arteries. Additionally, their cardiovascular risk factors, medical history, and medication use were evaluated, and a thorough cardiovascular examination was conducted by a cardiologist. It is important to note that this study has received approval from the Ethics Committee of Shandong Provincial Hospital, which is affiliated with Shandong First Medical University (SWYX: NO.2023-110). Furthermore, all recruited participants provided written informed consent prior to joining the study.

## Neuropsychological examinations

All participants in this study underwent cognitive assessments using both the MoCA and the Mini-Mental State Examination (MMSE) scales to evaluate their cognitive status.

The MoCA scale is a comprehensive assessment tool that evaluates different cognitive functions. These functions include visuospatial/executive abilities, naming skills, attention span, language proficiency, abstraction capabilities, memory recall, and orientation to time and place. By assessing these areas, the MoCA scale provides a holistic view of an individual's cognitive abilities and helps identify any potential CI or deficits.

The assessment of participants was conducted in a quiet environment following standardized procedures. The maximum score for both scales is 30 points. A total score below 26 points on the MoCA assessment was considered indicative of poor cognitive function, while a total score below 24 points on the MMSE assessment was considered indicative of poor cognitive function.

## Magnetic resonance imaging protocol

Whole-brain images were acquired at the Shandong Provincial Hospital Affiliated to Shandong First Medical University using a Siemens 3.0 T Prisma MR system and a 64-channel head coil for brain scanning. All participants were positioned inside the machine, with foam padding placed around their heads to minimize movement. They were instructed to remain still and keep their eyes closed during the imaging process.

During the resting-state fMRI scan, an echo-planar imaging sequence was used to capture blood oxygen level-dependent (BOLD) signals with the following parameters: repetition time (TR) = 2,000 ms, echo time (TE) = 30 ms, field of view (FOV) = 220 × 220 mm$^2$, in-plane resolution = 64 × 64, thickness/gap = 3.5/0 mm, flip angle = 90°, 33 axial slices. T1-weighted whole brain magnetization-prepared rapid gradient echo (MPRAGE) images were collected to capture anatomical details using the following parameters: TR = 2,530 ms, TE = 2.98 ms, TI = 1,100 ms, FOV = 256 × 256 mm$^2$, in-plane resolution = 256 × 256 mm$^2$, flip angle = 7°, and 192 axial slices. During the entire scan session, participants were instructed to maintain a relaxed state and refrain from thinking about anything in particular.

## Data preprocessing

The DPABI toolbox, which is based on SPM12, was used to preprocess all images (*Yan et al., 2016*). The preprocessing was performed using the MATLAB 2020 platform. The preprocessing pipeline included the following steps: (1) The initial 10 functional volumes were excluded to ensure signal stabilization. (2) The interleaved slice acquisitions in the functional data were corrected. (3) Head motion correction to exclude images with head motion >0.5 mm or rotation >0.5°, (4) Registration of the functional images to the MPRAGE anatomical image using the Montreal Neurological Institute (MNI) template, followed by spatial normalization. (5) Linear detrending. (6) Spatial smoothing using a 6 mm × 6 mm × 6 mm Gaussian smoothing kernel, full-width half maximum. (7) Filtering: performed bandpass filtering of 0.01–0.08 was performed to remove low-frequency drifts and high-frequency noise. The preprocessing steps of DC did not include smoothing beforehand.

## Calculation of DC

DC is a fundamental approach in assessing the significance of connections within a brain network and identifying key nodes within it *Li et al., (2016)*. To perform this analysis, the following steps are typically taken: 1. Whole-brain voxel-wise functional connectivity analysis was conducted on the preprocessed images. 2. Calculate correlation matrix: The time courses of each voxel in the subjects are correlated with those of all other voxels, resulting in a correlation matrix. To obtain the whole-brain correlation matrix, the Pearson correlation coefficient was calculated between each voxel and the remaining voxels in the brain. 3. Calculate voxel-wise degree of connectivity: The voxel-wise degree of connectivity is determined by summing the significant connections at the individual level. A correlation threshold, such as 0.25, is set to define which connections are considered significant (*Buckner et al., 2009*). 4. Standardize correlation matrix: The individual correlation matrix is standardized using Fisher's z-transform. 5. Spatial smoothing: A Gaussian kernel with a full width at half maximum (FWHM) of 6 mm was applied to perform spatial smoothing. 6. Group-level comparisons: Group-level comparisons of the degree centrality maps are conducted using a general linear model (GLM) analysis, often implemented in the DPABI toolbox. 7. Statistical analysis: Two-tailed two-sample t-tests are performed to observe the differences in degree centrality values between each pair of groups. The voxel-level and cluster-level threshold is typically set to $p < 0.001$. 8. Correction for multiple comparisons: AlphaSim correction was applied to the results to control for false positives.

The post-correction threshold is typically set to $p < 0.05$, with a cluster size threshold of ≥19 voxels (*Ledberg, Åkerman & Roland, 1998*).

## Dynamic casual modeling

Based on the findings of the DC analysis, brain regions exhibiting significant changes in DC are identified as regions of interests (ROIs). These ROIs are then analyzed to determine the directionality and strength of the functional connections between them. This involved extracting the time series data from these regions and assessing changes in directionality and connectivity strength using an optimized model. SpDCM analyses were performed using Statistical Parametric Mapping (SPM12, http://www.fil.ion.ucl.ac.uk/spm), a widely-used software package for neuroimaging data analysis.

The time series data of specific brain areas were averaged for each subject. This averaging process was based on the outcomes of the DC analysis. During the analysis of resting-state fMRI data, a generalized linear model (GLM) analysis was conducted in a fully connected mode. This mode enabled the identification of bi-directional connections between each pair of ROIs within each subject. Importantly, it is worth noting that no external input was introduced to the model throughout the analysis process. In this study, the EC between a specific set of ROIs was calculated using $2^4$ free parameters. These parameters were utilized to quantify the strength and directionality of the connections between the ROIs.

To determine the most appropriate dynamic causal model for the two groups, a *post hoc* model selection process was employed (*Friston & Penny, 2011*). This involved fitting the full model with all free parameters and conducting a "greedy search" resulting in 256

reduced models (*Rosa, Friston & Penny, 2012*). Subsequently, a *post hoc* model optimization process was performed using the posterior probability as a criterion to identify the most suitable model for each group. After identifying the optimized model, the parameters (DCM.Ep.A) were compared using one-sample t tests and two-sample t tests within each group in SPSS 20.0 software. A significance level of $p < 0.05$ was considered statistically significant. These rigorous statistical analyses provided insights into the most appropriate dynamic causal models for each group and highlighted the differences in parameter inference between and within the groups.

## Statistical analysis

In the present study, statistical analysis was conducted using SPSS 25.0. A two-sample t test was performed on the continuous variables in the clinical data of the subjects, including age, years of education, MoCA and MMSE scores. Additionally, a chi-square test was employed for comparing categorical data such as gender, hypertension, diabetes, and dyslipidemia. The significance level was set at 0.05. To evaluate the correlation between the strength of effective connectivity and clinical features of CAD, multivariate linear regression was employed while adjusting for confounding factors such as age, sex, hypertension, diabetes, and dyslipidemia. Statistical significance was defined as a $p$-value less than 0.05.

## RESULTS

We included 24 CAD patients and 24 HC with mean ages of $58.75 \pm 9.64$ and $56.87 \pm 12.18$ years, respectively. Among the 24 CAD patients, 12 (52.9%) were male, 12 individuals had a prior medical history of hypertension, 11 had a prior medical history of diabetes, 15 had a prior medical history of dyslipidemia, and 10 had a prior medical history of smoking. Table 1 shows the demographic and clinical information of the subjects. Compared to the healthy control group, CAD patients had significantly lower MMSE scores ($p < 0.05$) and significantly lower MoCA scores ($p < 0.001$). In addition, the scores for visuospatial/ executive ability, attention, language ability, and memory ability on the MoCA scale were significantly lower.

### Comparison of DC values between CAD and HC

After applying appropriate adjustments, the statistical analysis revealed significant differences in the DC values of the left parahippocampal cortex (PHC), the right superior frontal gyrus (SFG), and the left medial temporal gyrus (MTG) between the two groups (Fig. 1). Specifically, compared to the HC group, the CAD group had increased DC values in the right SFG and decreased DC values in the left PHC and left MTG. The results can be found in Table 2.

### Dynamic causal modeling analysis

In Figs. 2 and 3, our analysis has uncovered notable disparities in the strength of effective connectivity between individuals diagnosed with CAD patients and those in the HC. In comparison to the control group, the positive connectivity strength between the PHC and SFG, as well as between the PHC and MTG, was significantly reduced ($p < 0.05$) in

**Table 1 Demographics and clinical data.**

| Variable | CAD (*n* = 24) | HC (*n* = 24) | *p*-value |
|---|---|---|---|
| Demographic information | | | |
| Male sex, No. (%) | 12 (50.0%) | 15 (62.5%) | 0.334 |
| Age (years), mean (SD) | 58.75 ± 9.640 | 56.87 ± 12.181 | 0.982 |
| Hypertension, No. | 12 | 16 | 0.224 |
| Diabetes mellitus, No. | 11 | 6 | 0.259 |
| Dyslipidemia, No. | 15 | 11 | 0.741 |
| Smokers, No. | 10 | 13 | 0.292 |
| Education, years | 12.57 ± 3.627 | 12.53 ± 3.852 | 0.976 |
| MMSE score | 24.81 ± 2.562 | 27.29 ± 1.978 | <0.05 |
| MoCA score | 21.86 ± 4.211 | 26.80 ± 1.897 | <0.001 |
| Visuospatial/executive | 3.33 ± 1.278 | 4.79 ± 0.426 | <0.001 |
| Naming | 2.67 ± 0.796 | 2.93 ± 0.267 | 0.175 |
| Attention | 2.76 ± 0.436 | 3.00 ± 0.001 | <0.05 |
| Language | 1.81 ± 0.602 | 2.79 ± 0.426 | <0.001 |
| Abstraction | 1.38 ± 0.740 | 2 ± 0.001 | 0.619 |
| Memory | 2.52 ± 1.965 | 4.43 ± 1.505 | <0.05 |
| Orientation | 6.00 ± 0 | 6.00 ± 0 | |

**Note:**

MMSE, Mini-Mental State Examination; MoCA, Montreal Cognitive Assessment.

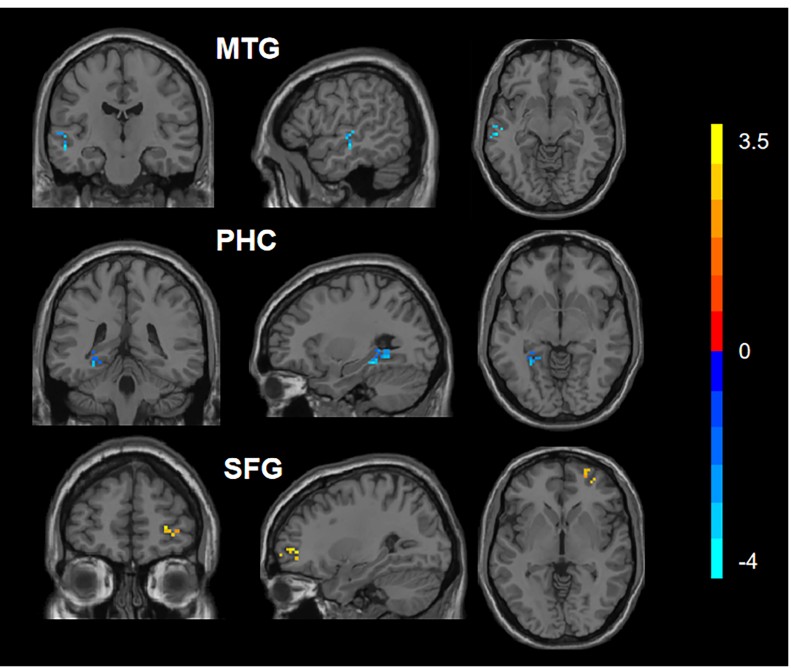

**Figure 1 Comparison of DC values between CAD and HC.** Blue represents the decreased DC values in the left parahippocampal cortex (PHC_L) and left medial temporal gyrus (MTG_L). Red represents increased DC values in the right superior frontal gyrus (SFG_R).

**Table 2 Brain regions with significant differences in DC values between the two groups.**

| Regions | Hemisphere | MNI coordinates | Voxel | Peak T value | p value |
|---|---|---|---|---|---|
| SFG | R | 27 48 −6 | 19 | 3.69569 | <0.001 |
| PHC | L | −30 −36 −12 | 31 | −4.51232 | <0.001 |
| MTG | L | −54 −18 −9 | 26 | −4.01634 | <0.001 |

Note:
MNI, Montreal Neurological Institute; L, left; R, right; SFG, Superior Frontal Gyrus; PHC, Parahippocampal cortex; MTG, Medial Temporal Gyrus.

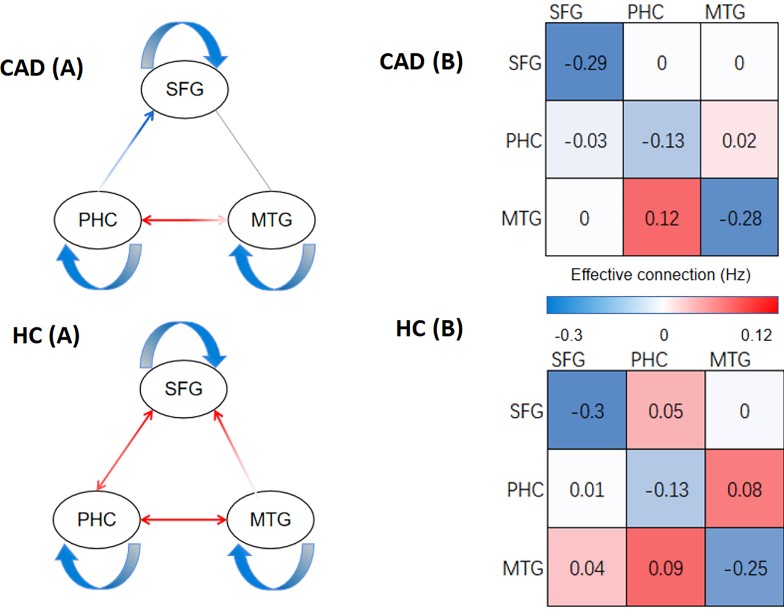

**Figure 2 The result of DCM of CAD and HC.** (A) The direction of effective connections ($P < 0.05$) between the CAD and HC groups, with positive connections shown in red and negative connections shown in blue. (B) The strength of effective connections (matrix read from left to right, $P < 0.05$) between the CAD and HC groups. PHC, parahippocampal cortex; SFG, superior frontal gyrus; MTG, medial temporal gyrus.

CAD patients. At the same time, we observed an enhancement of negative connectivity between the PHC and SFG. Furthermore, we found that the positive connectivity strength between the MTG and SFG disappeared in CAD patients, and the positive connectivity strength between the SFG and PHC was not detected.

To explore the correlation between patients' clinical scale and properties of effective connectivity, we conducted a Pearson's correlation analysis between the MoCA scores, MMSE scores and the strength of effective connectivity. This analysis aimed to shed light on the potential relationship between cognitive abilities, as assessed by the MoCA and MMSE, and the communication patterns within the brain networks. By examining the strength of effective connectivity, we can uncover potential associations between cognitive performance and the functional connectivity of the brain. Notably, we observed a significant positive correlation between the strength of connectivity from the PHC to the MTG and the MoCA score in CAD patients ($P < 0.05$). This finding suggests that stronger

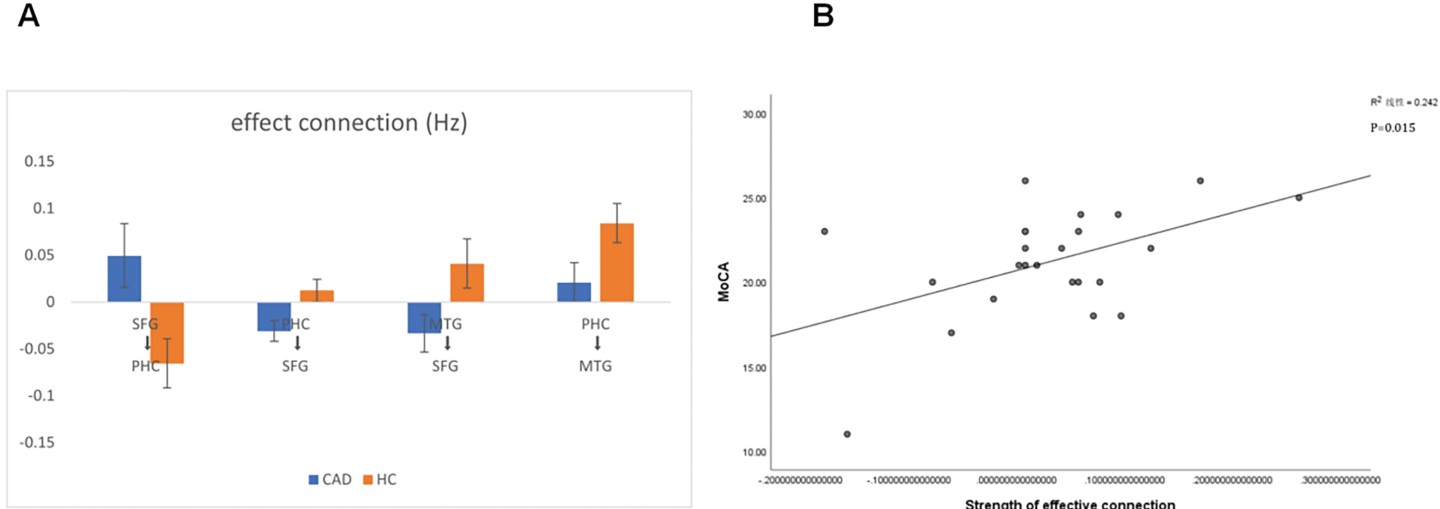

**Figure 3 Significant changes in effective connections (Hz) in the CAD patients.** (A) Mean effective connections of significantly changed pairs of regions. (B) The effective connections between PHC and MTG were positively correlated with the MoCA score. PHC, parahippocampal cortex; SFG, superior frontal gyrus; MTG, medial temporal gyrus.                               

connectivity between the PHC and MTG may be associated with better cognitive performance in CAD patients (Fig. 3B).

## DISCUSSION

CAD is a prevalent chronic condition that affects populations worldwide and is associated with a multitude of complications. CI is one of the complications of CAD, and research has demonstrated that CAD is an independent risk factor for cognitive impairment (*Balbaid et al., 2020*). Cerebral microangiopathy is strongly correlated with CAD, suggesting that it may indicate the presence of widespread cerebral atherosclerosis and cerebrovascular disease (*Wang et al., 2022*). Coronary atherosclerosis is a chronic disease. Long-term effects of cardiovascular risk factors as well as atherosclerosis on the cerebral vasculature cause chronic diffuse cerebral ischemia (*Gorelick et al., 2011*). The coronary atherosclerosis was strongly associated with lacunar infarction, cerebral white matter hyperintensities (WMH), and intracranial arterial stenosis. It has been shown that WMH patients have significantly reduced cerebral blood flow and significantly increased oxygen uptake fraction (*Meng, Yang & Jin, 2022*), while abnormal functional connectivity (FC) in the brain of WMH patients has been found, which may be associated with cognitive impairment (*Wang et al., 2020*).

Many studies have proven that the cognitive function of CAD patients is obviously decreased, including attention, memory, language ability, and executive function (*Ibáñez & Fernández-Alvira, 2019*; *Okumiya et al., 2007*). Our research findings also indicate that patients with CAD exhibit significant declines in visuospatial/executive abilities, attention span, language proficiency, and memory recall compared to HC. This is consistent with previous research findings. CAD is currently considered to affect brain function in many ways. This may be due to ischemic damage and cerebral microvascular disease caused by

CAD. In addition, CAD may also affect cognitive function through mechanisms such as inducing inflammatory responses and activating the coagulation system (*Li et al., 2011*).

To the best of our knowledge, this is the first study to use spDCM to analyze changes in effective connectivity between brain regions in the CAD population. In this study, we first analyzed the DC values of the brain functional connectivity network and found significant differences in the DC values of CAD patients in the SFG, PHC, and MTG compared to the HC. This is similar to previous research results and supports the relationship between cognitive dysfunction and abnormal brain functional connectivity (*Niu et al., 2022*). Specifically, we found that the DC values of the PHC and MTG in CAD patients decreased, suggesting that the abnormal functional connectivity of these brain regions may be related to CI. The PHC, as the core area of the brain's internal memory system, is closely related to multiple aspects of cognitive function, including spatial memory, episodic memory, and visual recognition (*Eichenbaum, Yonelinas & Ranganath, 2007*). Research has found that patients with Alzheimer's disease and mild CI exhibit decreased regional homogeneity in the parahippocampal gyrus, which is correlated with CI (*Zhang et al., 2012*). Other research has shown that patients with mild CI have decreased functional connectivity in the parahippocampal gyrus, which is related to CI (*Liang et al., 2011*).

Our neuropsychological examinations results also showed that the overall cognitive ability of CAD patients, as well as memory and visuospatial/executive aspects, significantly decreased, which may be related to the decreased DC values of PHC and other connectivity strengths. The MTG is considered a key area of the brain's internal language processing and memory and is closely related to cognitive functions such as language comprehension and semantic processing (*Hickok & Poeppel, 2007*). *Luo et al. (2022)* studied resting-state functional magnetic resonance imaging (fMRI) in patients with amnestic mild cognitive impairment (aMCI). The results showed that the MTG DC value of aMCI patients decreased, which may be related to impairments in language comprehension and semantic memory function (*Luo et al., 2022*). Meanwhile, other research has explored changes in brain networks in patients with Alzheimer's disease (AD). The study found that the DC value of the MTG in AD patients decreased, which may be related to CI in memory, attention, and executive function (*Agosta et al., 2012*). Our results showed that the DC value of the MTG in CAD patients significantly decreased compared to that in HC, and CAD patients also showed a decline in language function in neuropsychological examinations, which is consistent with previous research results. In summary, the results of this study support the important role of the PHC and MTG in cognitive function.

On the other hand, we also found that the DC value of the SFG in CAD patients increased. The SFG is part of the prefrontal cortex and is closely related to advanced cognitive functions such as decision-making, planning, and execution (*Koechlin, Ody & Kouneiher, 2003*). We believe that this may be due to self-compensation, that is, while the patient's cognitive function declines, the functional connectivity of the SFG region is enhanced to compensate for the functional deficiencies of other brain regions (*Wassermann et al., 2010*).

## Differences in connectivity between the two groups

Our study differs from previous research in that we used spDCM to evaluate the differences in effective brain connectivity between two groups based on DC, thereby combining the strengths of both approaches. This enabled us not only to assess the static characteristics of brain networks and the importance of each brain node between the two groups but also to establish dynamic causal models between different brain regions, providing a more comprehensive analysis of the relationships between brain regions. As academic researchers, we believe that this approach offers a more sophisticated and nuanced understanding of differences in brain connectivity. In our study, we utilized resting-state fMRI analysis with spDCM to investigate connectivity patterns in patients with CAD compared to controls. By examining the strength of connections, interpreted as rate constants of neuronal responses, we aimed to identify specific effective connectivity and their directionality in the resting-state fMRI data.

The PHC is a very important region in the brain, which is a critical structure for the function of the hippocampus and plays a role in higher cognitive functions. The connections between the PHC and other brain regions are important for overall cognitive function, memory function, visual recognition, and more (*Eichenbaum, Yonelinas & Ranganath, 2007*). Therefore, we can infer that when the effective connections between the PHC and other brain regions decrease, the balance between brain regions may be disrupted, leading to a decline in cognitive ability in CAD patients. Our study found that the information flow between the PHC and MTG in CAD patients was blocked, which may lead to impaired cognitive function. *Milton et al. (2012)* used fMRI to investigate brain activity during an autobiographical memory task in the transient epileptic amnesia (TEA) population. The study found that the effective connectivity between the PHC and the MTG was significantly decreased in the TEA group compared to the HC, indicating that a decrease in effective connectivity between the PHC and MTG may lead to a decline in memory and overall cognitive ability in patients (*Milton et al., 2012*). *Xia, Fu & Qian (2016)* used resting-state fMRI to investigate the role of PHC functional connectivity changes in memory impairment after concussion. The study found that the functional connectivity between the PHC and MTG was significantly decreased in patients with memory impairment after concussion compared to healthy controls, and the study also found that the degree of functional connectivity decrease was significantly correlated with the severity of memory impairment. In addition, the study found that the connectivity changes between the parahippocampal gyrus and the temporal lobe after concussion may be related to neural network reorganization in the brain. In our correlation analysis, we also found that the decrease in effective connectivity between the PHC and MTG in CAD patients was significantly correlated with the severity of CI. This indicates that the decrease in effective connectivity between the PHC and MTG has an important impact on the decline in cognitive function in CAD patients.

The SFG is a key node in both the default mode network (DMN) and the cognitive control network (CEN), and changes in its effective connectivity with other brain regions may significantly affect cognitive function. Our study showed that CAD patients had

significantly lower effective connectivity strength between the SFG and PHC compared to HC, and the connection direction changed from positive in HC to negative in CAD patients. This suggests that SFG positively stimulates PHC to complete cognitive tasks differently in CAD patients compared to HC, and the connectivity between SFG and PHC is inhibited in CAD patients, which may be an important reason for the decline in cognitive function in CAD patients. *Peng & Burwell (2021)* explored the importance of PHC and SFG in regulating context-modulated behavior and emphasized the importance of transmission between SFG and PHC in context-modulated behavior, formation of memories, emotion regulation, and decision making. They proposed that the connection between SFG and PHC may be more important than the connection between the hippocampus and parahippocampal gyrus (*Peng & Burwell, 2021*). This suggests that the decrease in effective connectivity between SFG and PHC may have serious effects on cognitive function. In our results, we also found that the limited connectivity strength between MTG and SFG decreased in CAD patients, and the connection direction changed from positive in HC to negative in CAD patients. MTG is a crucial brain region, and *Xu et al. (2019)* employed resting-state fMRI and coactivation pattern analysis to investigate the functional characteristics of the temporal gyrus. By utilizing these techniques, they aimed to gain a deeper understanding of the temporal gyrus and its role in brain function. They found that the temporal gyrus plays different roles in language, memory, and social cognition, and there are multiple subregions within the temporal gyrus that are closely related to cognitive functions, such as functional connectivity (*Xu et al., 2019*). In terms of neural fiber connectivity, *Briggs et al. (2021)* used diffusion imaging to study the potential fiber bundle structure of MTG. The results showed that MTG is highly involved in the brain's language networks and higher-order cerebral networks such as the DMN (*Briggs et al., 2021*). In terms of brain network connectivity, *Lu et al. (2022)* found that the functional connection between the MTG and the SFG is closely related to individual thought inhibition, emphasizing the importance of the SFG and the MTG. The functional connection between these two regions may play an important role in controlling negative emotions and inhibiting thoughts (*Lu et al., 2022*). These studies, similar to our results, reveal the neuroimaging mechanism underlying changes in cognitive function by measuring changes in effective connectivity between different brain regions in CAD patients. They highlight the important role that changes in effective connectivity strength and direction between the SFG, PHC, and MTG can play in altering cognitive function.

## Limitation

Our study does have certain limitations that should be acknowledged. One of the main limitations is the relatively small sample size, which consisted of only 24 patients with CAD and 24 HC participants. The small sample size may limit the generalizability of our findings and reduce the statistical power of our analyses. There may be limitations in exploring the effective connectivity of brain networks between the two groups, and our study subjects were recruited from a single medical institution, which may have recruiting bias. For example, the study may not have included all populations with CAD. Second, in terms of neuropsychological examination, we only tested the MMSE and MoCA scales for

the subjects and evaluated the cognitive function of the subjects based on the scores of the two scales. In the future, we will increase the number of scales to more specifically evaluate the degree of decline in various cognitive functions. At the same time, this study only considered the differences in network connections between CAD patients and HC, but there may be other potential factors that could affect the research results, such as drug treatments and individual differences among patients. In future studies, we will further increase the number of subjects and optimize the experimental process in order to achieve more accurate experimental results and more discoveries.

## CONCLUSIONS

Overall, by combining DC and spDCM methods in resting-state fMRI, we found that there are brain functional network disorders in patients with CAD. In this study, it was observed that CAD patients exhibited a significant decrease in the DC values within the left PHC and left MTG compared to the HC. Conversely, there was an increase in DC values within the right SFG in CAD patients compared to HC. Based on the differences in DC values in different brain regions, we conducted effective connectivity analysis and found that the positive connection between the SFG and PHC disappeared, the connection strength weakened, the negative connection between the PHC, MTG and SFG increased, the connection strength between MTG and SFG decreased, the positive connection between PHC and MTG weakened, and the connection strength decreased. Moreover, the decrease in effective connectivity strength between the OHC and MTG was closely related to cognitive impairments. This study indicates that there are disruptions in the brain functional network structure of CAD patients, which may contribute to a deeper understanding of the neuropathophysiological mechanisms underlying CI in CAD.

### Funding
The authors received no funding for this work.

### Competing Interests
The authors declare that they have no competing interests.

### Author Contributions
- Rui Qin performed the experiments, analyzed the data, prepared figures and/or tables, and approved the final draft.
- Tong Li conceived and designed the experiments, authored or reviewed drafts of the article, and approved the final draft.
- Cuicui Li performed the experiments, analyzed the data, prepared figures and/or tables, and approved the final draft.
- Lin Li performed the experiments, prepared figures and/or tables, and approved the final draft.

- Ximing Wang conceived and designed the experiments, authored or reviewed drafts of the article, and approved the final draft.
- Li Wang conceived and designed the experiments, authored or reviewed drafts of the article, and approved the final draft.

### Human Ethics

The following information was supplied relating to ethical approvals (*i.e.*, approving body and any reference numbers):

Ethics Committee of biomedical research involving people in Shandong Provincial Hospital(Ethical Application Ref: swyx:no. 2023-110).

### Data Availability

The data is available at figshare: Qin, Rui (2023). Rawdata.7z. figshare. Dataset. https://doi.org/10.6084/m9.figshare.23703582.v1.

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
