# Peer review of "Investigating altered brain functional hubs and causal connectivity in coronary artery disease with cognitive impairment"

_PeerJ, doi:10.7717/peerj.16408_

## Round 0.1 · original submission · Major Revisions

The authors raised several major concerns. The authors should carefully address them.

Reviewer 1 ·

Basic reporting

1. In the “BACKGROUND” section of the abstract, the second sentence describes the methodology of the manuscript. The author should state why they investigated the resting-state connectivity and effective connectivity. Could probing brain connectivity provide a biological explanation for the disease? Or can it provide guidance for clinical strategies? Or other.

2. For the use of abbreviations, the abbreviation is indicated where it first appears in the main text, followed by the abbreviation name. And then you can use the abbreviated name instead of writing the full name.

3. In the “Dynamic Casual Modeling” section of the methodology, it is unclear why the author selected the superior frontal gyrus (SFG), posterior hippocampus (PHC), and middle temporal gyrus (MTG). If this is a hypothetical study (with a priori areas of interest), then in the introduction you should provide a background statement about these three areas, including why you are focusing on them, how they have changed in other relevant studies, and how these areas may change in your study.

4. The legend of Figure 2 was unclear. The author should explain what the color represents, and specify the direction of effective connection in the right figure.

5. Correct “eccective” to “effective” in Figure 3.

6. The association between the strength of connectivity from the posterior hippocampus to the middle temporal gyrus and the MoCA score in patients may not be significant correlated. The P-value was not corrected by multiple correction. I think the P-value was not survived after the Bonferroni correction.

Experimental design

In materials, part of the patients had a history of smoking. Can these participants be allowed to smoke before the scan, and what is the interval between smoking and the start of the scan? I suggested the author specified the details about the scanning.

Validity of the findings

This study included 24 CAD patients and 24 healthy controls to explore the brain connectivity. It is a relatively small sample for functional MRI study. How the author confirms the generalization or stability of the findings.

Additional comments

no comment.

Reviewer 2 ·

Basic reporting

1. Delete line 10 "dddd" and line 12 "1*".
2. The font of line 13 needs to be changed.
3. A study needs to be changed into "This study ".
4. The authors need to explain that this study is a prospective or retrospective study?
5. line 23 "Coronary Artery Disease (CAD)" into "CAD" , and please check all abbreviations yourself and correct the whole manuscript, such as line 46, 69, 79, 91, 94 and so on.
6. "CAD patients" needs to be corrected into "patients with CAD" in line 28
7. " Through cognitive assessments," is a wrong sentence in line 30 .
8. The manuscript still needs to be proofread carefully for English grammar errors, because there are several spelling and grammatical errors, especially in Materials & Methods.
9. "58.75±9.640 and 56.87±12.181" changes into "58.75±9.64 and 56.87±12.18" in line 217-218.
10. fig legends can not be shown in a screenshot of the figure.

Experimental design

1.why did you only use the MOCA scores in line 243-253?
2.There is a lack of sufficient description of the relationship between cardiovascular disease and cognitive impairment in the entire text, and more emphasis is placed on the relationship between cognitive impairment and brain regions

Validity of the findings

no comment

Additional comments

Revise the section of discussion, a paragraph to discuss the relationship between CAD and cognitive impairment is needed.

·

Basic reporting

no comment

Experimental design

no comment

Validity of the findings

This study investigated altered brain functional hubs and causal connectivity in coronary artery disease with cognitive impairment using the degree centrality and the dynamic causal model. However, the following problems need to be further considered:
1.Does the data preprocessing involve the covariate regression process? Are the global signals removed?
2. For cluster-level correction, the initial threshold should be set at p<0.001. Otherwise, it will increase the probability of false positive results.
3. Why are there 23 free parameters in the dynamic causal model?
4. Please provide the p value(mean value or peak value) for each region in Table 2 and the respective means of two groups. Similarly, P values also need to be provided in Figure 3.
5.There is spelling mistake in Figure 3B, please check for similar mistakes in the whole manuscript.
6. As shown in Figure 2, it might be more readable to label the values directly in the graph.

Additional comments

no comment

---

## Round 0.2 · Major Revisions

The authors should carefully consider the reviewers' concerns.

Reviewer 1 ·

Basic reporting

1. The figures does not match the manuscripts.

2. “Response: We are sorry for not providing sufficient explanation for the selection of Regions of Interest (ROI). Prior to the section on "Dynamic Casual Modeling," we first conducted DC analysis on subjects (CAD, HC) to identify brain regions with differential DC values between the two groups. These regions (SFG, PHC, MTG) were then used as seed points for spDCM analysis to obtain effective brain connectivity between SFG, PHC, and MTG. It is not a hypothetical study. We have provided further explanation on this section in the manuscript.” I don't think you should name specific brain areas in the methods section, I suggest you state is as "The brain regions with significant changes in DC are defined as regions of interest (ROI) for analysis in this section. Specific brain region names should appear in the results section.

3. With respect to abbreviations, the main text part and the abstract part are separate. Abbreviations used in the main text should state the full name where they first appeared.

Experimental design

no

Validity of the findings

no

Reviewer 2 ·

Basic reporting

1.The abbreviation in this article still has issues; the abbreviation should start from the introduction.The abbreviation 'MRI' on line 191 is still incorrect; it is written as 'MRI' on line 97. So check it again please.
2. The author changed from 23 to 24 on line 359, but used superscript. It is recommended to correct this.
3.There are errors in the figure legends and tables ; firstly, I saw in Figure 1 that the caption is 'fig2,' 'fig3' in Figure 2, and 'fig1' in Figure 3. Additionally, all the text is still included in the article as screenshots. I would like to remind you to make corrections.
4.Some of the reference documents need correction. For example, some references contain anomalies or lack information, some are missing DOIs, and there are references to incorrect documents. It is recommended to make replacements accordingly.
“Armstrong NM, Bangen KJ, Rhoda A, and Gross AL. Associations Between Midlife (but Not Late-Life) Elevated Coronary Heart Disease Risk and Lower Cognitive Performance: Results From the Framingham Offspring Study. American Journal of Epidemiology:12. ”
“Balbaid NT, Al-Dawalibi A, Khattab AM, Al-Saqr F, and Bashir S. 2020. The Relationship between Cognitive Impairment and Coronary Artery Disease in Middle-aged Adults. Cureus 12. ”
“Camici PG, and Filippo C. 2014. Coronary microvascular dysfunction. Springer.
Eichenbaum H, Yonelinas AP, and Ranganath C. 2007. The medial temporal lobe and recognition memory. Annual Review of Neuroscience 30:123-152. ”
“Furie K, Kasner S, Adams R, Albers G, Bush R, Fagan S, Halperin JL, Johnston SC, Katzan I, and Kernan WN. 2014. American Heart Association Stroke Council, Council on Cardiovascular Nursing, Council on Clinical Cardiology, and Interdisciplinary Council on Quality of Care and Outcomes Research (2011) Guidelines for the prevention of stroke in patients with stroke or transient ischemic attack: a guideline for healthcare professionals from the American Heart Association/American Stroke Association. ”
“Lo K, Liu Q, Madsen T, Rapp S, and Liu S. 2019. Relations of magnesium intake to cognitive impairment and dementia among participants in the Women's Health Initiative Memory Study: a prospective cohort study. BMJ Open 9:e030052. ”
“Ledberg A, Akerman S, and Roland PE. 1998. Estimation of the probabilities of 3D clusters in functional brain images. NeuroImage 8:113-128. ”
"Wu J, Lu AD, Zhang LP, Zuo YX, and Jia YP. 2019. [Study of clinical outcome and prognosis in pediatric core binding factor-acute myeloid leukemia]. Zhonghua Xue Ye Xue Za Zhi = Zhonghua Xueyexue Zazhi 40:52-57. 10.3760/cma.j.issn.0253-2727.2019.01.010"
5.I recommend verifying the results with strict correction using the Bonferroni correction.

Experimental design

no comment

Validity of the findings

While you have provided answers to my question(“Revise the section of discussion, a paragraph to discuss the relationship between CAD and cognitive impairment is needed.”) and added some information, I would advise you to delve a bit deeper into the connections, such as how certain physiological changes in CAD can lead to reduced blood flow and result in cognitive impairments.

Additional comments

no comment

·

Basic reporting

no comment

Experimental design

no comment

Validity of the findings

no comment

Additional comments

no comment

---

## Round 0.3 · accepted · Accept

This manuscript can be accepted now.

Reviewer 1 ·

Basic reporting

NO

Experimental design

NO

Validity of the findings

NO

Reviewer 2 ·

Basic reporting

no comment

Experimental design

no comment

Validity of the findings

no comment

Additional comments

no comment